# Men’s Positive and Negative Experiences Following Acute Myocardial Infarction

**DOI:** 10.3390/ijerph18031053

**Published:** 2021-01-25

**Authors:** MCarmen Solano-Ruiz, Genival Fernandes de Freitas, M. Idoia Ugarte-Gurrutxaga, Sagrario Gómez-Cantarino, José Siles-González

**Affiliations:** 1Department of Nursing, University of Alicante, 03690 San Vicente, Spain; jose.siles@ua.es; 2Escola de Enfermagem, Universidade de São Paulo, 05508-220 Sao Paulo, Brazil; genivalf@usp.br; 3Department of Nursing, Physiotherapy and Occupational Therapy, University of Castilla-La Mancha Campus, 45004 Toledo, Spain; Maria.Ugarte@uclm.es (M.I.U.-G.); sagrario.gomez@uclm.es (S.G.-C.)

**Keywords:** myocardial infarction, rehabilitation, activity of daily living, qualitative research, social phenomenological

## Abstract

(1) Objective: To describe men’s experiences as acute myocardial infarction sufferers from a social phenomenological perspective, a year after the event (2) Methods: The phenomenological interview was used to capture the participants’ discourse. The data were analyzed according to the theoretical methodological approach of social phenomenology. (3) Results: The discourse analysis of the content produced the following categories, set out according reasons “why”: personal biography, knowledge set, warning signs prior to the illness, experience at the intensive care unit, and rehabilitation process; and reasons “for”: expectations as regards the illness, health professionals, and future social life and work prospects. (4) Conclusions: Participants had not established a healthy condition one year after myocardial infarction, perceiving a very thin line between life and death. Personal biography influences the coping of the disease. They feel like the illness helped them to create new meanings and value of life. They envisage a future full of great restrictions and uncertainty. The results of this study have underlined the need to involve care at all stages of the illness: the physical and emotional dependence upon admittance at the intensive care unit, the need to be cured, the constant demand for information about the illness, the difficulties encountered upon returning home, uncertainty about the future, etc. All these moments indicate that proper nursing care adapted to the specific needs of each individual and their family members must be provided in order to help them to overcome all the stages involved in this process. It is necessary to individualize care because the sense of reality is common and universal, but the ways of expressing are subjective, and it depended on the totality of experiences accumulated throughout life.

## 1. Introduction

Heart disease has remained the leading cause of death at the global level for the last 20 years. However, it is now killing more people than ever before. The number of deaths from heart disease has increased by more than 2 million since 2000. The European region has seen a relative decline in heart disease, with deaths falling by 15% [1,2]. In Spain, cardiovascular disease is the main cause of hospitalization and death [3], although evidence of a drop in mortality over the last two decades [4] and a rise in morbidity [5] clearly indicates that the lethal nature of the disease has diminished.

Individuals suffering from a heart disease, such as myocardial infarction (MI), may undergo major changes in their lives. Several research works have involved longitudinal studies about the perception of the illness over time [6,7,8]. Many men described a series of losses after the cardiac diagnosis; these include loss of physical strength, emotional health, depression, fatigue, and problems with physical functioning [9,10]. In the immediate aftermath of the MI, patients restructured and re-evaluated their attitudes towards self, life, religious beliefs, and others [7,11,12].

This study forms part of the field of phenomenology as a structure of reference and, in keeping with our main aim, is based on a social phenomenological approach that will enable us to discover and understand the situation that people experience after suffering a heart attack.

One study attempted to incorporate the phenomenological method into the social sciences based on the architecture established by Husserl, proposing an organized knowledge of social reality as a primary objective [13].

For social phenomenology, people live in the world in natural attitude. They have the ability to intervene in this world naturally, influencing and being influenced, transforming themselves continuously and changing social structures [13].

The aim was to understand the world in relation to others in the realm of intersubjectivity, analyzing social relationships as mutual relationships that involve people. The fundamental purpose is to understand certain social actions that have a contextualized effect on shaping the social, and not just individual aspect.

Action is interpreted by the subject from their existential motive, derived from the experiences recorded in subjectivity, constituting leading wires of the action in the social world. Those relating to the achievement of objectives, expectations, and projects are called “reasons for”, and those based on history, body of knowledge, and experience within the biopsychological context of the person are called “reasons why”. The set of reasons for and why refers to typical situations with means and purposes [14].

The biography of people who suffered a myocardial infarction occupies the space of social action, characterized by the baggage of available knowledge that is constructed through previously accumulated experiences and which characterizes the typifications of the world, generated through social life.

Therefore, we understand that part of existential motivation has both a meaning that is subjective (experienced by subjects) and objective (referring to a situation which shows itself significant for those who experience the phenomenon investigated).

Various research related to the field of health has applied the social phenomenological approach in order to gain an insight into the phenomenon [15,16].

The aim of this study was to describe the men’s experiences as acute myocardial infarction sufferers from a social phenomenological perspective, a year after the event.

## 2. Materials and Methods

Qualitative and phenomenological research was chosen for this study, conceived in the light of social phenomenology [13]. This approach permitted the investigation of this social group of men who experience a given typical situation.

The participants were chosen using purposeful sampling in accordance with the maximum variation sampling strategy [17]. The total number of people who participated in the study was 14.

The following criteria were applied when selecting participants for the study: (1) men who had received a medical diagnosis of acute myocardial infarction, (2) those who had been admitted to the intensive care unit within the previous year, (3) those who were able to communicate properly, excluding those who suffered sensory impairments, (4) those currently living at home and diagnosed as stable as regards their illness, (5) patients who had suffered a relapse of their illness were included in the study, provided that this was not the first time they had suffered this pathology.

A total of 20 participants were contacted, 4 of them declined to participate in the study and 1 had difficulty hearing. The other 15 participants met the inclusion criteria, one of whom died days before the interview.

This study was carried out in the Region of Valencia (SE Spain), in Health Department number 20. The reference of this department is the General University Hospital of Elche, which has twelve beds for intensive care.

After receiving the relevant permission from the Departmental Management Office and approval for the study from the Clinical Research Ethics Committee, we accessed the database of patients admitted to the intensive care unit, thus preselecting those subjects who met the inclusion criteria. We then contacted each of the preselected subjects by telephone to ask for their voluntary collaboration in the study, explaining our aim and informing them that the data would be collected from them at their home during an interview that would deal with issues related to their life and illness.

There, participants gave their authorization and signed the informed consent, guaranteeing therefore the ethical research principles in accordance to the Law 02/2016 of Biomedical Researches.

Data collection was carried out from June to December 2017.

Confidentiality was guaranteed in the processing and use of any data supplied was exclusively for the purposes established in this research. In order to obtain an overall view of each participant, the semi-structured interview was chosen as the most appropriate tool for capturing the true essence of the process experienced since the onset of the disease (a myocardial infarction) until reaching a certain degree of stability, i.e., until the patient was at home without any of the symptoms associated with myocardial infarction. The interview began with general questions such as: “Could you tell something about the experience of your illness?”, “How would you describe your experience at the intensive care unit?”, “What has suffering this illness meant to you?”, and “What changes have happened in your life as a result of this illness?” Other questions complemented the interview (Table 1).

The researchers were able to ask for clarification of unclear answers at any time throughout the interview. Interviews were digitally recorded and later transcribed for attentive reading of the statements.

As the interview progressed, themes that had emerged throughout the discourse were discussed in further detail.

To ensure the anonymity of participants in this study, they were identified as follows: D1, D2, D3, etc., corresponding to Discourse 1, Discourse 2, Discourse 3, and so on.

The data analysis was carried out using the steps proposed by Parga Nina and other researchers of social phenomenology. These steps were the following [15,16,17]:

(a) Reading of discourse in order to discover the experiences lived by the participants, first with view to identifying the global sense of these. (b) Re-reading of the transcripts, identifying specific categories that express significant aspects related to the participants’ experiences as regards in-order-to and because motives. (c) Grouping of meaning units extracted from the discourse that represent a point of convergence within the content, creating identified categories. (d) Establishing the meanings of social action based on the participants’ discourse in order to obtain the typical experience.

After gathering the convergences that emerged from interviews, the appearing categories were warning signs prior to the illness, origin of the illness, fear of dying in the intensive care unit, association of the disease with family and friends, expectations as regards their illness, expectations as regards health professionals, and social expectations. That information was discussed according to the framework developed for social phenomenology, in addition to literature in experiences of life following of myocardial infarction field to enrich it.

## 3. Results

### 3.1. Participants

The total number of people who participated in the study was 14 men. Age ranged between 41 and 66 years (mean = 54.7). Only one of the participants was single and did not have children, seven were married and five separated and all of these participants have children (mean = 2.2). As regards education, nine had completed primary education, four had completed secondary education, and only one had further education qualifications. All except one of the participants, who was a pensioner, were employed at the time of their illness, which had since changed to pensioner for four of them, five were still on sick leave, and only four had gone back to work. Most of the participants (10) had a middle socioeconomic status and the rest low (Table 2).

The convergence of the reasons why and the reasons for, emerging from the analyzed statements, permitted capturing the meanings that patients gave to having a heart attack after one year. Figure 1 shows the main findings that correspond to the explanatory model.

### 3.2. Reasons “Why” of the Myocardial Infarction Sufferers

Warning signs prior to the illness: most of the participants claimed to have had some complaint prior to the illness but had not paid too much attention to it or had blamed it on some other cause.

“*A month before, I walked out of work, I couldn’t stand it there any more, I just couldn’t, I sat down, I knelt, as if I was out of it, I lay on my back. Then I blamed it on stress and it never occurred to me that it might have been a heart attack ….*”(D13)

Origin of the illness: the participants attributed the origin of their illness to work-related stress and an excessive workload throughout their lives.

“*Everything is coming out now, I’ve worked a lot, I’ve had to work in the fields and the suchlike since I was 11, I’ve worked so much my whole life, so very, very much*”(D3)

They claimed to have had an extremely busy social life, accompanied by an excessive consumption of alcohol and tobacco; factors which they believe favored the appearance of their illness.

“*My life in general was really crazy, I mean, I’ve always liked partying although I don’t think I partied to excess and I’ve always liked know going out and disappearing for two or three days, and making the party last all weekend*”(D2)

All participants except one claimed to have been heavy smokers and were aware of the consequences.

“*I smoked since I was 12 or 13 until last year*”. “*I ate a lot of salt, a lot of fat. I never cared*”(D12)

Fear of dying in the intensive care unit: the people interviewed expressed fear and anxiety when admitted to the intensive care unit and lived that period as if close to death. Those who had previously been admitted to the unit felt calm and drowsy under the effects of sedation.

“*When I was in the ICU I cried a lot because I was afraid that I was already dead*”(D4)

Association of the disease with family and friends: they knew of the disease as family members or friends had been afflicted and consequently died.

“*I knew someone who had a heart attack and it’s like I say sometimes, (…) it’s precisely those who have had a heart attack that die from one*”(D5)

### 3.3. Reasons “For” of the Myocardial Infarction Sufferers

Expectations as regards their illness and reordering goals: all participants were in a state of anticipation in relation to their illness; they stated that they were afraid of becoming disabled and that they may have another attack that will lead to their death.

“*When I got home, I was afraid and we’ve been back to the hospital twice*”(D14)

“*Before myocardial infarction, I had decided to change my 100 m^2^ home to 150 m^2^, or change my car, after, I thought about it. I said to myself that I am all running around for a better life, if I am to be bed-ridden, does it really make any difference if my home is 100 or 150 m^2^ or if my car is a Peugeot or Elegance Mercedes*”(D12)

Expectations as regards health professionals: most of the participants commented on the lack of information provided by health professionals. Although the participants were greatly reassured by the way the health professionals had acted during an emergency situation, they would expect more humane care and treatment so that unnecessary invasive techniques may be avoided.

“*I was given that information in the corridor and I… I was really scared, it’s not that I don’t want more information but I thought that they should be the ones to tell me how I should live my life from now on*”(D7)

Social expectations: all participants had reduced their social activities mainly due to limited physical activity, lack of confidence when driving, changes in eating habits, and having given up smoking and alcohol consumption. They expressed the need to spend more time with their loved ones; giving great importance to being surrounded by those who care.

“*Now I drive less, before we didn’t used to go out a lot but we did go away at Christmas*”(D1)

Work prospects: Only four participants had gone back to work, at the same post they had held prior to their illness; four were classed as unable to work through disability and the rest were on temporary sick leave.

“*I get very tired, the stairs kill me and if not them, the hills, I can’t manage the hills, I’m a pensioner now*”(D6)

Reflections on life, weighing up the lifecycle: The illness has marked a turning point in the participants’ lives. At the time of the interview (one year after the event), all participants claimed to have taken stock of their lives.

“*Life has changed me a lot. Life has no meaning when you are sick, let alone having a painful heart. You will value life when you get sick, even if you have the world, it will be valueless*”(D5)

## 4. Discussion

The results from our study show two clearly distinct ways of perceiving the same phenomenon: acute myocardial infarction. These two main categories are “positive and negative coping experiences” [7]. On the one hand, there are those who consider it a threatening disease that interrupts their daily life, generating negative feelings and disapproval towards it. This illness is marked by a loss of spontaneity in actions and lack of energy, which means adapting oneself to an unhealthy heart [11,12,13,14,15,16,17,18,19,20,21,22]. In contrast, the other group is made up of people who treat this phenomenon as a warning sign, a chance to change their lifestyle, and who view it in a positive light [22], patients restructured and re-evaluated their attitudes towards self, life, religious beliefs, and others [7].

Admittance to the intensive care unit causes participants to become scared about death, making them feel that they are very close to the end. Fear and anxiety are two key emotional reactions that accompany the patient during their stay in the ICU [12,23]. The patients took into consideration the MI as a warning that made them aware of the very thin line between life and death [21]. During their stay, the sufferers positively rate the professionalism with which the health workers act, the reassurance provided by both the staff and the environment, as well as the quick and efficient manner with which emergency situations are dealt [24]. Despite good interpersonal relationships, the participants expect health workers to be able to transmit appropriate information about the process to the patients. Poor communication not only caused anxiety while the patient was in ICU but also contributed to less than optimal recoveries after discharge. The participants in the study expressed the need for a more humane treatment that would prevent unnecessary invasive techniques or waiting times when faced with an emergency situation.

The importance of family support as a source of affection and the unique link with the outside world has been made obvious. The need to feel supported, the demand of constant company, the security and sense of being provided with calm by their loved ones as well as the importance of support firstly, from their partner, and secondly, from their children [23,24] have been some of the points set forth by those who have participated in our study.

Most participants in the study had not had direct prior contact with the illness, despite stating that they had experienced certain symptoms related to ischemic cardiomyopathy. They consider themselves to be very healthy and do not perceive any risk factors associated with this disease. Patients tend to ignore or reinterpret symptoms through rationalization and denial of their vulnerability to this illness [25]. On the whole, participants attribute the origin and cause of their illness to work stress and an excessive workload throughout their lives [12]. They considered smoking to be a secondary risk factor. Although they do not consider eating habits to be associated to risk factors, they do realize that their diet is not adequate. These results coincide with some authors’ statements, which say that patients had their own view of the onset of disease. The patients spent much time and energy considering the causes and seeking explanations for their disease [12].

Just like those who had previous heart problems, attributed the appearance of the illness to the fact that they had not looked after themselves properly. This generated a sense of guilt [6,25].

The weakness and fatigue experienced during the rehabilitation process hinders everyday activities. The global fatigue was associated with concurrent symptoms, such as breathlessness and stress, and coping strategies such as changes in values, intrusion, and isolation. The patients felt emotionally and physically powerless and described their life as “transfixed” [6,7].

Poor physical health and low levels of physical activity after myocardial infarction affect negatively when returning to work [26]. It is very difficult for men, who considered themselves the family breadwinners [6]. They felt that they had let down their wives and experienced guilt. Our results show that inability to work often forces people to take early retirement or, in some cases, to change to a job that requires less physical effort.

Due to the sufferers’ lack of confidence about driving their cars, their leisure activities are limited. Consequently, this stage is perceived as a turning point—a time for changing lifestyles, applying restrictions, and steering life into a new phase [21,26]. This means assuming the responsibility necessary to bring about such changes, the search for professional help, and how to take care of one’s self in the future [6,7,8,9,10,11,12,13,14,15,16,17,18,19,20,21].

Health education of patients by nursing professionals has been proposed as the main intervention tool in order to encourage them to adopt lifestyle changes. Healthcare professionals and patients do not always have the same priorities regarding what information is most important to consider. The patients declared that they would have needed clearer guidelines concerning the rehabilitation process [7].

The participants in the study are currently living in a state of expectation as regards their illness [7]. Being constantly on guard means that sufferers will tend to visit their health center to check any sign they may have detected, which in most cases bears no relation to their coronary disease. Being afraid of having another heart attack is the biggest concern sufferers have during this stage of the illness [21]. They envisage a future full of great restrictions and uncertainty.

They are worried about resuming sexual activity as, although they are aware that there is no need to restrict themselves in this sense, they are concerned about how to act in this situation and they express the need to be informed accordingly. When people express concern, decreasing frequency, and less desire, this is usually due to a lack of knowledge or ability to look for alternatives regarding the new conditions imposed by the illness [27].

Different authors [28] affirm the need to provide telephone help lines to facilitate monitoring and provision of the necessary patient information. An integrative review of the research suggests that people with cardiac disease showed some benefits from nurse-led/delivered telephone interventions [29].

The illness stabilization phase involves a process of introspection and looking back over the whole life cycle. New values and incentives appear to stimulate change: they appreciate time spent at home, feeling the need to enjoy their family as much as possible, and state that they are living a more organized and quiet life [6,11]. Some authors concluded that patients realize that they have the chance to live a new and better life by reconsidering their daily existence. They considered the illness a positive event which encourages a change in physical and eating habits which in turn reaps benefits [6,9].

We can therefore state that those who have had a heart attack have passed a resilience survival test, characterized by the ability of an individual or social system to live well and to develop positively despite difficult living conditions and, furthermore, to come out stronger and transformed as a result [24]. In combination with their experience of going through a key life transition, these younger people also search for a new personal meaning in life [14]. To sum up, they have been able to recognize the illness itself and have strengthened their own ability to face up to the serious problems caused by the illness and, from here on, make the most out of life [8].

Most of the patients interviewed for this study mentioned their wife as the main source of support throughout all the stages of the illness. On the other hand, spouses are often aware that everything is not in order and may become overprotective inadvertently increasing the patients’ sense of inferiority or feeling that they are not relied on anymore [7,8,9,10,11,12,13,14,15,16,17,18,19,20,21,22,23,24,25,26,27,28,29,30]. They acted as his guardian by phoning home when at work, being available, being on the alert, being attentive to the patient, and being prepared [21,22,23,24,25,26,27,28,29,30].

An important detail discovered in our study was the support perceived by those people who lived in an environment where there were small children, as they stated that the latter were the main source of help in overcoming the process as they believed they must continue living for the children.

The application of phenomenological sociology theory [13] has allowed us to understand a way of thinking, reasoning, and acting of people who suffer from AMI. This theory’s objective is to focus in the social relations in the world of life. People are immersed in a social, historical, and cultural context. Health care professionals must consider the amount of knowledge and experience acquired over a lifetime, as well as biographical situation, in order to help themselves understand the lived phenomenon.

The persons, during their life, see the world from the perspective of their own interests, motives, desires, ideology, and religious commitments. The reality of common sense is culturally considered universal, however, the way these forms are expressed in individual life depends on the totality of experience that the subject constructs in the course of his concrete existence.

### 4.1. Study Implications

This paper has important implications for the team of professionals that make up the cardiology unit. The results of this study have underlined the need to pay attention on all stages of the illness: the physical and emotional dependence upon admittance at the intensive care unit, the need to be cured, the constant demand for information about the illness, the difficulties encountered upon returning home, uncertainty about the future, etc. All these moments indicate that proper nursing care adapted to the specific needs of each individual and their family members must be provided in order to help them to overcome all the stages involved in this process.

It is necessary to provide a new structure that would allow the development of patient-centered and family interventions after AMI that embrace patients’ preferences, needs, values, and goals, that are tailored to each stage of each person’s recovery.

### 4.2. Limitations

This study was carried out in the national and local context of the Spanish society and its results should be interpreted cautiously due to its qualitative nature of study. The generalization of our data to other countries such as the United States can be complicated by cultural differences between the two countries.

The main limitation of this study is given by the composition of the sample. Future research should investigate the perception of the disease in people with a high socioeconomic profile or with university studies. It would be interesting to expand our findings with women and establish gender differences with respect to the disease.

## 5. Conclusions

The results of this study show the different situations experienced by people who have suffered a heart attack. A period of one year after MI is considered by scientists as a stability moment, but our findings show us how people feel that their lives drastically changed. The participants regarded the MI as a warning that made them aware of the very thin line between life and death. They feel like the illness helped them to create new meanings and value of life. They envisage a future full of great restrictions and uncertainty.

It can be concluded that in spite of problems and stresses experienced by patients, myocardial infarction may also have positive effects for them. That is, patients may draw some positive experiences from their illness.

## Figures and Tables

**Figure 1 ijerph-18-01053-f001:**
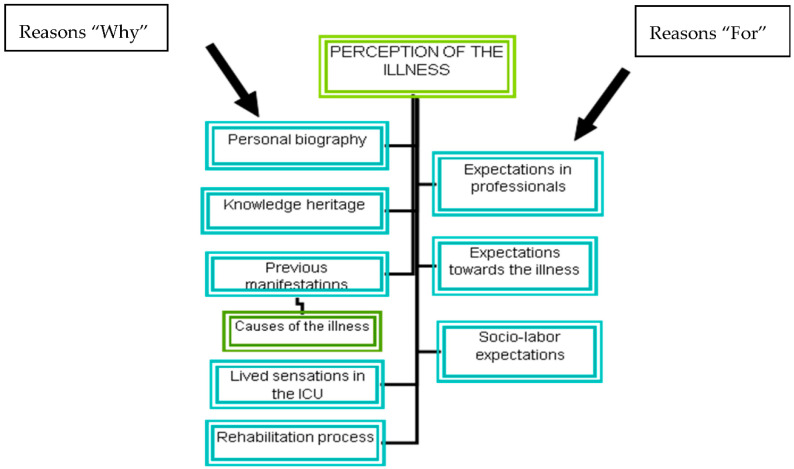
Analysis of the content according reasons “why” and reasons “for”.

**Table 1 ijerph-18-01053-t001:** Interview guide.

Prior Illnesses
Antecedents, risk factors
Diagnosis, dates, therapies
Prior hospital admissions
Social and family support
Beliefs (religion…)
Current
Changes in lifestyle, work, sexuality, partnerships
The emotional situation
Feelings and meaning of the illness.

**Table 2 ijerph-18-01053-t002:** Sociodemographic description of participants.

Participants	Age	Marital Status	Current Employment Status	Level of Education	Socio-Economic Level	Prior Record of Illnesses
D-1	61	married	Pensioner	Primary	Mid-high	No priors
D-2	50	married	Active	Primary	Mid	* AMI-2008
D-3	58	married	Pensioner	Primary	Mid	** IDDM
D-4	62	married	Active	Secondary	Mid	No priors
D-5	60	separate	Unemployed	Secondary	Mid-low	No priors
D-6	41	single	Pensioner	Primary	Low	AMI 2014 chest pain
D-7	52	married	Active	Secondary	Mid	AMI 2000
D-8	46	separate	Active	Higher	Mid-high	No priors
D-9	57	separate	Active	Primary	Low	High blood pressure
D-10	66	married	Pensioner	Higher	Mid	Int. Claudication
D-11	61	separate	Sick leave	Primary	Low	Prior infarctions
D-12	53	married	Sick leave	Primary	Low	No priors
D-13	46	separate	Sick leave	Primary	Low	No priors
D-14	54	married	Sick leave	Primary	Low	No priors

* AMI: acute myocardial infarction. ** IDDM: insulin dependent diabetes mellitus.

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
