# Peer review of "Men’s Positive and Negative Experiences Following Acute Myocardial Infarction"

_ijerph, 2021, doi:10.3390/ijerph18031053_

Round 1
Reviewer 1 Report
All comments and suggestions have been responses correctly by authors
Author Response
No comments from the reviewers
Reviewer 2 Report
I have read with pleasure the manuscript from Solano-Ruiz et al.: "Positive and negative men’s experiences following acute myocardial infarction".
Although the sample size is quite small, as the authors highlighted in the limitation section, the article is well-written.
I did not understand why part of the text is in red (track-changes like), since I think it is the first round of revisions.
The authors should check the manuscript for some typos, e.g.:
- line 334: check spaces
- line 350: It would be interesting to It would be interesting to increase
- we findings -> our findings
- please reword the conclusion section: this sentence "This study shows the men ’ experiences of acute myocardial infarction sufferers" does not make sense to this reviewer.
Author Response
Point 1: Although the sample size is quite small, as the authors highlighted in the limitation section, the article is well-written.
R: We agree with your suggestion, in future studies we will increase our sample.
Point 2: I did not understand why part of the text is in red (track-changes like), since I think it is the first round of revisions.
R: Part of the text is in red because it is the second round of revision
Point 3: line 334: check spaces
R: Spaces have been removed in this sentence
Point 4: line 350: It would be interesting to It would be interesting to increase
we findings -> our findings
R: The change was made in accordance with the proposal
Point 5: please reword the conclusion section: this sentence "This study shows the men ’ experiences of acute myocardial infarction sufferers" does not make
R: This sentence has been changed to:
The results of this study show the different situations experienced by people who have suffered a acute myocardial infarction.
This manuscript is a resubmission of an earlier submission. The following is a list of the peer review reports and author responses from that submission.
Round 1
Reviewer 1 Report
Ref: ijerph-981751
Title: Positive and negative experiences following acute 3 myocardial infarction
The manuscript “Positive and negative experiences following acute myocardial infarction” by Solano-Ruiz et al. aimed to describe the men’s experiences of acute myocardial infarction sufferers from a social phenomenological perspective, a year after the event.
The study includes very limited population, considering that acute myocardial infarction is one the most common pathologies during cardiologists’ daily practice.
Women were excluded from the study because as authors state “the number of women with myocardial infarction would be very low”. Myocardial infarction is not rare amongst females and they may have very different approach respect to males.
Since there are only 14 patients included authors not compared patients with first and recurrent events
The results of the study due to previous observations are not generalizable.
Reviewer 2 Report
An interesting study is presented to understand men's experiences following acute myocardial infarction. The paper needs minor revisions, which are detailed below:
Title
- It should be clarified that the study was conducted on men, e.g.: "Positive and negative men's experiences following acute ...".
Abstract:
- Apply recommendation on implications for practice, included in the discussion section of this review.
Introduction
- It is suggested that data on the MI be incorporated not only in Spain, but also in the rest of the world, especially at European countries, as a minimum.
- The authors describe in detail the theoretical framework of the study, based on social phenomenology. I believe that this explanation could be reduced and an effort made to link the interest that this theory has in the objective of this research.
Materials and Methods
- The sampling process should be described in more detail. How many subjects were contacted at the outset?, how many met the inclusion criterio?, and how many were interviewed?. It could also be discussed whether the sample is representative or there are some profiles that could be further explored in future research. Last question can be addressed as limitation, into the discusión section.
- Semi-structured interview: In addition to the general questions, the other questions used should be included. It is suggested that a table with the interview guide be included.
Results
- Participants: In addition to the table, it is suggested that a paragraph be included highlighting the overall profile of the sample, based on the criteria described in the table.
- The participants' quotations should be put in italics, to differentiate it from the authors' descriptions or interpretations.
- New text should be included to explain Figure 1. It follows that this is the explanatory model that summarizes the findings of the study, but requires additional information to be understood.
Discussion
- Authors should include a section on limitations at the end of the discussion. Although some issues are deduced throughout the study, a specific part should be devoted to this.
Conclusions
- Implications for practice: Why only nursing professionals?, do authors not believe that these findings can involve more professionals?. I think this issue should be discussed further.